# Host ecology regulates interspecies recombination in bacteria of the genus *Campylobacter*

**Evangelos Mourkas**[1], **Koji Yahara**[2], **Sion C Bayliss**[1], **Jessica K Calland**[1], **Håkan Johansson**[3], **Leonardos Mageiros**[1], **Zilia Y Muñoz-Ramirez**[4], **Grant Futcher**[1], **Guillaume Méric**[1], **Matthew D Hitchings**[5], **Santiago Sandoval-Motta**[4], **Javier Torres**[4], **Keith A Jolley**[6], **Martin CJ Maiden**[6], **Patrik Ellström**[7], **Jonas Waldenström**[3], **Ben Pascoe**[1,8], **Samuel K Sheppard**[1,6]*

[1]The Milner Centre for Evolution, Department of Biology and Biochemistry, University of Bath, Bath, United Kingdom; [2]Antimicrobial Resistance Research Center, National Institute of Infectious Diseases, Tokyo, Japan; [3]Centre for Ecology and Evolution in Microbial Model Systems, Linnaeus University, Kalmar, Sweden; [4]Unidad de Investigacion en Enfermedades Infecciosas, UMAE Pediatria, Instituto Mexicano del Seguro Social, Mexico City, Mexico; [5]Swansea University Medical School, Swansea University, Swansea, United Kingdom; [6]Department of Zoology, University of Oxford, Oxford, United Kingdom; [7]Department of Medical Sciences, Zoonosis Science Centre, Uppsala University, Uppsala, Sweden; [8]Faculty of Veterinary Medicine, Chiang Mai University, Chiang Mai, Thailand

*For correspondence:
s.k.sheppard@bath.ac.uk

Competing interest: The authors declare that no competing interests exist.

**Abstract** Horizontal gene transfer (HGT) can allow traits that have evolved in one bacterial species to transfer to another. This has potential to rapidly promote new adaptive trajectories such as zoonotic transfer or antimicrobial resistance. However, for this to occur requires gaps to align in barriers to recombination within a given time frame. Chief among these barriers is the physical separation of species with distinct ecologies in separate niches. Within the genus *Campylobacter*, there are species with divergent ecologies, from rarely isolated single-host specialists to multihost generalist species that are among the most common global causes of human bacterial gastroenteritis. Here, by characterizing these contrasting ecologies, we can quantify HGT among sympatric and allopatric species in natural populations. Analyzing recipient and donor population ancestry among genomes from 30 *Campylobacter* species, we show that cohabitation in the same host can lead to a six-fold increase in HGT between species. This accounts for up to 30% of all SNPs within a given species and identifies highly recombinogenic genes with functions including host adaptation and antimicrobial resistance. As described in some animal and plant species, ecological factors are a major evolutionary force for speciation in bacteria and changes to the host landscape can promote partial convergence of distinct species through HGT.

## Editor's evaluation

This article will be of broad interest to readers who work in bacterial genomics, particularly those conducting research on *Campylobacter* genomics. This article substantially advances the field by quantifying horizontal gene transfer among sympatric and allopatric species in natural populations, and demonstrating enhanced horizontal gene transfer between *Campylobacter* species that colonize the same host species.

**Figure 1.** Barriers to horizontal gene transfer (HGT) in bacteria. A series of barriers must be surmounted for DNA to transmit from one species to another. These are broadly defined in three categories. At a given time, alignment of holes in successive barriers is necessary for HGT to occur. Here, we focus on ecological barriers that are influenced by multiple factors that reflect the physical isolation of bacteria in separate niches.

## Introduction

It is well established that bacteria do not conform to a strict clonal model of reproduction but engage in regular horizontal gene transfer (HGT) (*Smith et al., 1991*). This lateral exchange of DNA can confer new functionality on recipient genomes, potentially promoting novel adaptive trajectories such as colonization of a new host or the emergence of pathogenicity (*Sheppard et al., 2018*). In some cases, gene flow can occur at such magnitude, even between different species (*Shapiro et al., 2016*; *Doolittle and Zhaxybayeva, 2009*), that one may question why disparate lineages do not merge and why distinct bacterial species exist at all (*Doolittle and Papke, 2006*). An answer to this lies in considering the successive processes that enable genes from one strain to establish in an entirely new genetic background.

The probability of HGT is governed by the interaction of multiple factors, including exposure to DNA, the susceptibility of the recipient genome to DNA uptake, and the impact of recombined DNA on the recipient strain. These factors can be broadly defined in three functional phases, and HGT can only occur when gaps align in each successive ecological, mechanistic, and adaptive barrier within a given time frame (*Figure 1*). In the first phase, the quantity of DNA available to recipient strains is determined by ecological factors such as the distribution, prevalence, and interactions of donor and recipient bacteria, as well as the capacity for free DNA to be disseminated among species/strains. In the second phase, there are mechanistic barriers to HGT imposed by the homology dependence of recombination (*Fraser et al., 2007*) or other factors promoting DNA specificity – such as restriction-modification, CRISPR interference, or antiphage systems (*Budroni et al., 2011*; *Oliveira et al., 2016*; *Doron et al., 2018*; *Nandi et al., 2015*; *Marraffini and Sontheimer, 2008*) – that can act as a defense against the uptake of foreign DNA (mechanistic barriers) (*Thomas and Nielsen, 2005*; *Eggleston et al., 1997*). Finally, the effect that HGT has on the fitness of the recipient cell in a given selective environment (adaptive barrier) will determine if the recombinant genotype survives for subsequent generations (*Sheppard et al., 2018*; *Zhu et al., 2001*).

Understanding how ecology maintains, and potentially confines, distinct strains and species has become increasingly important in the light of global challenges such as the emergence and spread of zoonotic pathogens (*Boni et al., 2020*). A typical approach to investigating this is to consider spill-over of particular strains or clones from one host to another (clonal transmission). This is an important phenomenon and may be influenced by anthropogenic change, such as habitat encroachment or agricultural intensification (*Mourkas et al., 2020*). However, in many cases, important phenotypes, including antimicrobial resistance (AMR) (*Johnson and Woodford, 2013*; *Schwarz and Johnson, 2016*; *Baker et al., 2018*), can be conferred by relatively few genes. In such cases, it may be important to consider how cohabiting strains and species can potentially draw genes from a common pange-nome pool (*Young, 2016*; *McInerney et al., 2020*; *Vos and Eyre-Walker, 2017*; *Werren, 2011*) and

how genes, rather than clones, can transition between segregated populations (gene pool transmission). To investigate the impact of ecological segregation (ecological barriers) on this gene pool transmission, in natural populations, requires quantification of HGT among sympatric and allopatric bacteria.

Species within the genus *Campylobacter* are an ideal subject for considering how ecology influences the maintenance of genetically distinct species for several reasons. First, *Campylobacter* are a common component of the commensal gut microbiota of reptiles (*Giacomelli and Piccirillo, 2014*; *Fitzgerald et al., 2014*), birds (*Griekspoor et al., 2013*; *Atterby et al., 2018*), and mammals (*Leatherbarrow et al., 2007*) but, being microaerophilic, do not survive well outside of the host. This creates island populations that have some degree of ecological isolation. Second, because at least 12 species have been identified as human pathogens (*Man, 2011*) and *C. jejuni* and *C. coli* are among the most common global causes of bacterial gastroenteritis (*Kaakoush et al., 2015*), large numbers of isolate genomes have been sequenced from potential reservoir hosts as part of public health source-tracking programs (*Sheppard et al., 2009a*; *Sheppard et al., 2009b*). Third, within the genus there are species and strains that inhabit one or multiple hosts (ecological specialists and generalists; *Mourkas et al., 2020*; *Griekspoor et al., 2013*; *Sheppard et al., 2011a*; *Dearlove et al., 2016*; *Sheppard et al., 2010*; *Sheppard et al., 2014*; *Woodcock et al., 2017*). As a single host can simultaneously carry multiple lineages (*Colles et al., 2008*), possibly occupying different sub-niches within that host (*Colles et al., 2015*), there is potential to compare allopatric and sympatric populations. Finally, high-magnitude interspecies admixture (introgression) between *C. jejuni* and *C. coli* isolated from agricultural animals suggests that host ecology plays a role in the maintenance of species (*Sheppard et al., 2013*; *Taylor et al., 2021*; *Sheppard et al., 2008*; *Sheppard et al., 2011b*).

Here, we quantify HGT among 600 genomes from 30 *Campylobacter* species using a 'chromosome painting' approach (*Thorell et al., 2017*; *Lawson et al., 2012*; *Yahara et al., 2013*) to characterize shared ancestry among donor and recipient populations. Specifically, we investigate the role of ecological barriers to interspecies gene flow. By identifying recombining species pairs within the same and different hosts, we can describe interactions where co-localization enhances gene flow, quantify the impact of ecological barriers in these populations, and distinguish highly recombinogenic genes that are found in multiple genetic backgrounds. This provides information about the evolutionary forces that give rise to species and the extent to which ecological barriers maintain them as discrete entities.

## Results

### Host-restricted and host-generalist *Campylobacter* species

Isolate genomes were taken from publicly available databases to represent diversity within the genus *Campylobacter,* including environmental isolates from the closely related *Arcobacter* and *Sulfurospirillum* species, to provide phylogenetic context within the *Campylobacteraceae* family (*Figure 2—figure supplement 1*). In total, there were 631 isolates from 30 different *Campylobacter* species (*Figure 2a*) and 64 different sources, isolated from 31 different countries between 1964 and 2016 (*Supplementary file 1*). Among the isolates, 361 were *C. jejuni* and *C. coli* and could be classified according to 31 clonal complexes (CCs) based upon sharing four or more alleles at seven housekeeping genes defined by multilocus sequence typing (MLST) (*Supplementary file 1*; *Dingle et al., 2001*) and were representative of known diversity in both species (*Mourkas et al., 2020*; *Sheppard et al., 2011a*). The obligate human commensal and pathogen *C. concisus* (n = 106 isolates) comprised two genomospecies (GSI, n = 32, and GSII, n = 74), as previously described (*Kirk et al., 2018*; *Supplementary file 1*). The collection also included 52 *C. fetus* isolate genomes, including three subspecies: *C. fetus* subsp. *fetus* (n = 8), *C. fetus* subsp. *venerealis* (n = 23), and *C. fetus* subsps. *testudinum* (n = 21) (*Supplementary file 1*; *Iraola et al., 2017*). Two clades were observed in *C. lari* (*Figure 2—figure supplement 2*), which could correspond to previously described subspecies based on 16S rRNA sequencing (*Debruyne et al., 2009*).

A maximum-likelihood phylogeny of the *Campylobacter* genus was reconstructed on a gene-by-gene concatenated sequence alignment of 820 gene families shared by >95% of all isolates, with a core genome of 903,753 base pairs (*Figure 2a*). The phylogeny included species that appear to be restricted to one host or environment, including: *C. iguanorium* (*Gilbert et al., 2015*) and *C.*

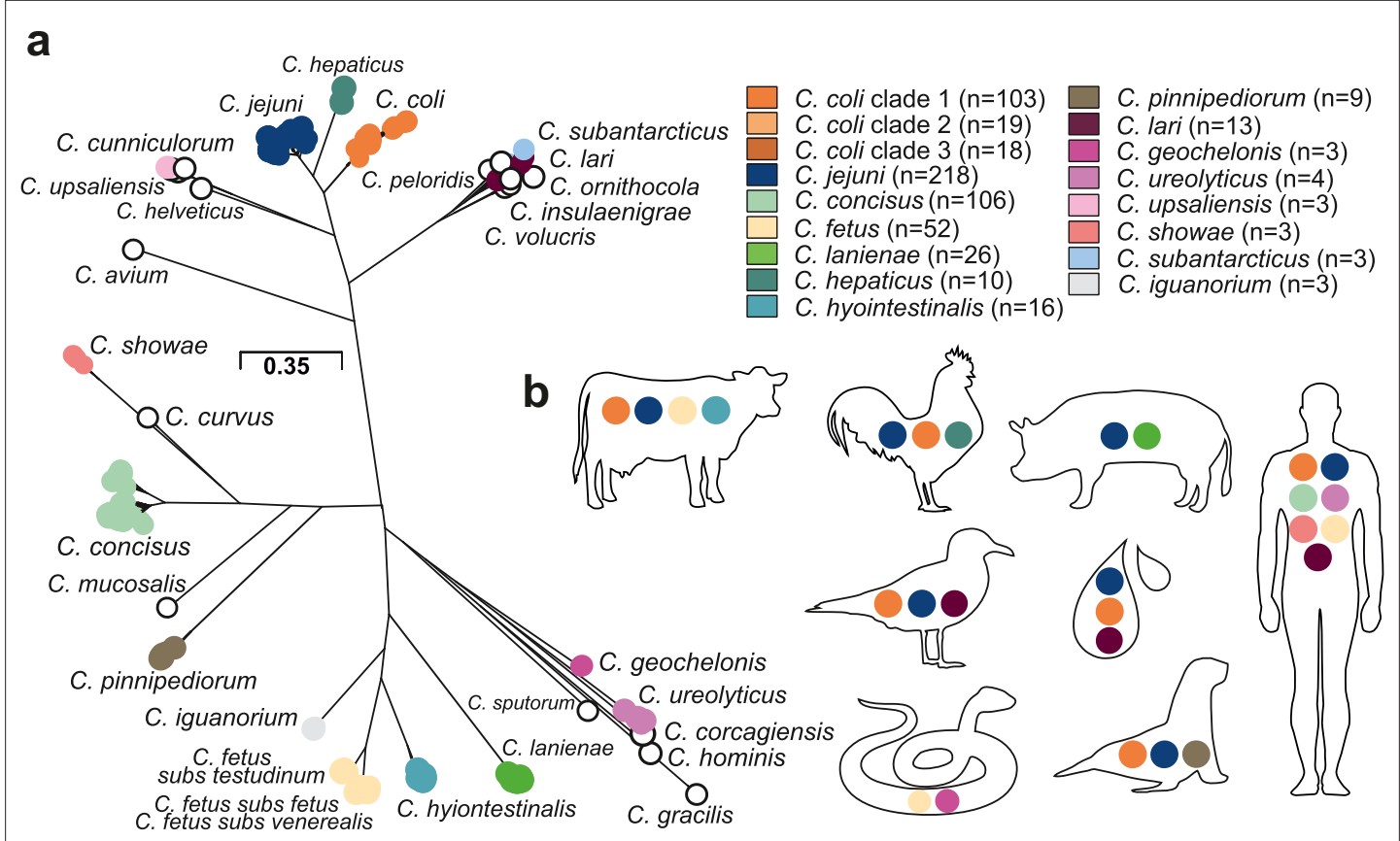

**Figure 2.** Population structure and host ecology in the genus *Campylobacter*. (**a**) Phylogenetic tree of 631 *Campylobacter* isolates from 30 species reconstructed using a gene-by-gene concatenated alignment of 820 core genes (shared by >95% of isolates) and an approximation of the maximum-likelihood (ML) algorithm implemented in RAxML. The species name is indicated adjacent to the associated sequence cluster. The scale bar indicates the estimated number of substitutions per site. (**b**) Isolation source of *Campylobacter* species with n ≥ 3 isolates.

The online version of this article includes the following figure supplement(s) for figure 2:

**Figure supplement 1.** Population structure of the *Campylobacteraceae* family.

**Figure supplement 2.** Core genome species trees.

**Figure supplement 3.** Overview of host associations of *Campylobacter* species.

**Figure supplement 4.** Core genome species trees.

geochelonis (**Piccirillo et al., 2016**) (reptiles); *C. lanienae* (**Logan et al., 2000**) (pigs); *C. hepaticus* (**Van et al., 2016**) (chicken liver); the *C. lari* group (**Miller et al., 2014**) (marine birds and environment); *C. pinnipediorum* (**Gilbert et al., 2017**) (seals) species - most of which were discovered recently (**Figure 2—figure supplement 3**). There was no evidence that phylogeography was reflected in the observed population structure for *Campylobacter* isolates from multiple hosts and countries (**Figure 2—figure supplement 4**). This is unsurprising as it is well known that host-associated genetic variation transcends phylogeographic structuring in *Campylobacter* (**Sheppard et al., 2010**). While some low-level local gene flow can be identified within a given country (**Pascoe et al., 2017**), this is vastly outweighed by recombination within particular host niches (**Sheppard et al., 2014**), particularly in small isolate collections such as those for some of the species in this study.

Host-restricted species had lower diversity possibly linked to low sample numbers, with *C. hepaticus* having the lowest diversity (**Figure 2—figure supplement 2**) with 8/10 genomes associated with isolates from the same outbreak (**Van et al., 2016**). For other species, there was evidence of a broad host range (ecological generalists) (**Figure 2b**). For example, highly structured *C. jejuni* and *C. coli* isolates were sampled from seven and six host sources, respectively (**Figure 2—figure supplement 2**, **Figure 2—figure supplement 3**, **Supplementary file 1**). For *C. fetus,* there was distinct separation between mammal-associated *C. fetus* subsp. *fetus* and *C. fetus* subsp. *venerealis*

and reptile-associated *C. fetus* subsp. *testudinum* (*Figure 2—figure supplement 2*) as previously described (*Iraola et al., 2017*). Unsurprisingly, a large proportion of the isolates in this study were from humans, likely reflecting intensive sampling. *C. jejuni* (27.52%; n = 60/218), *C. coli* (14.68%; n = 32/218), and *C. concisus* (44.5%; n = 97/218) were all common among human clinical samples. However, less common species were also present, with nearly half of all *Campylobacter* species (44.83%, n = 13/29) isolated from humans at least once (*Figure 2b*, *Supplementary file 1*). Agricultural animals were also a common source accounting for more than 1/3 of the isolates (38.35%; 242/631), with 10/30 *Campylobacter* species isolated from more than one source (*Figure 2b*, *Supplementary file 1*).

## Evidence of interspecies recombination in the core and accessory genome

Genome size varied between 1.40 and 2.51 Mb (*Figure 3—figure supplement 1*) (mean 1.73), and the number of genes (per isolate) ranged from 1,293 to 2,170 (mean 1,675) (*Figure 3—figure supplement 2*). The pangenome for the genus comprised 15,649 unique genes, found in at least one of the 631 isolates (*Figure 3—source data 1*), with 820 genes (5.24% of the pangenome) shared by >95% of all isolates (core genome), across 30 species (*Figure 3—source data 1*). We excluded species with fewer than three isolates in subsequent analysis. For the remaining 15 species, the core genome ranged in size from 1,116 genes in *C. lari* to 1,700 in *C. geochelonis* (*Figure 3a*, right panel, *Figure 3—source data 1*). Differences were also noted in the size of accessory genomes, with *C. concisus* (mean: 981 genes), *C. hyointestinalis* (mean: 946 genes), *C. showae* (mean: 1,160 genes), *C. geochelonis* (mean: 1,021 genes), and *C. fetus* (mean: 912 genes) containing the highest average number of accessory genes (*Figure 3a*, left panel, *Figure 3—source data 1*). Functional annotation of all 14,829 accessory genes showed that 71% (10,561) encoded hypothetical proteins of unknown function due to the lack of homology with well-characterized genes (*Figure 3—figure supplement 3*; *Pascoe et al., 2019*). Remaining genes were related to metabolism, DNA modification, transporters, virulence, inner membrane/periplasmic, adhesion, regulators, metal transport, and AMR (*Figure 3—figure supplement 3*).

To further understand genetic differentiation within and between species, we generated genus-wide similarity matrices for the core and accessory genomes (*Figure 3c and d*, *Figure 3—source data 1*). For the core genome, pairwise average nucleotide identity (ANI) was calculated for shared genes in all possible genome pairs (*Figure 3—source data 1*) using FastANI (*Jain et al., 2018*). On average, isolates of the same species shared >95% similarity (*Figure 3—source data 1*), with decreasing genetic similarity (between 85 and 90%) over greater phylogenetic distances. The number of core genome SNPs ranged from 983 to 230,264 for the 15 *Campylobacter* species with ≥3 isolates in our dataset, with *C. coli* and *C. concisus* having the greatest mean SNP numbers (*Figure 3—figure supplement 4a*), indicating considerable diversity within these species. In contrast, *C. hepaticus* and *C. geochelonis* had low mean SNP numbers with 986 and 4,310, respectively. This is likely related to low sample numbers with isolates either sampled in close proximity (*Piccirillo et al., 2016*) or from a single outbreak (*Van et al., 2016*).

The core genome similarity matrix provided initial evidence of interspecies gene flow (introgression). This can be observed as elevated nucleotide identity between *C. jejuni* and clade 1 *C. coli* (*Figure 3—source data 1*), consistent with previous studies (*Sheppard et al., 2013*; *Sheppard et al., 2008*; *Sheppard et al., 2011b*). Further evidence of introgression came from pairwise ANI comparison of genus-wide core genes, in all isolates of the 15 major *Campylobacter* species, to the *C. jejuni* genome (*Figure 3—figure supplement 4b*). In the absence of gene flow, isolates from the two species should have an approximately unimodal ANI distribution reflecting accumulation of mutations throughout the genome. This was largely the case, but for some species, low nucleotide divergence suggested recent recombination with *C. jejuni*. There was also evidence of interspecies accessory genome recombination. Presence/absence patterns in the accessory genome matrix show considerable accessory gene sharing among several species that was inconsistent with the phylogeny (*Figure 3—source data 1*). This is well illustrated in *C. lanienae* where much of the accessory genome was shared with other *Campylobacter* species (*Figure 3—source data 1*).

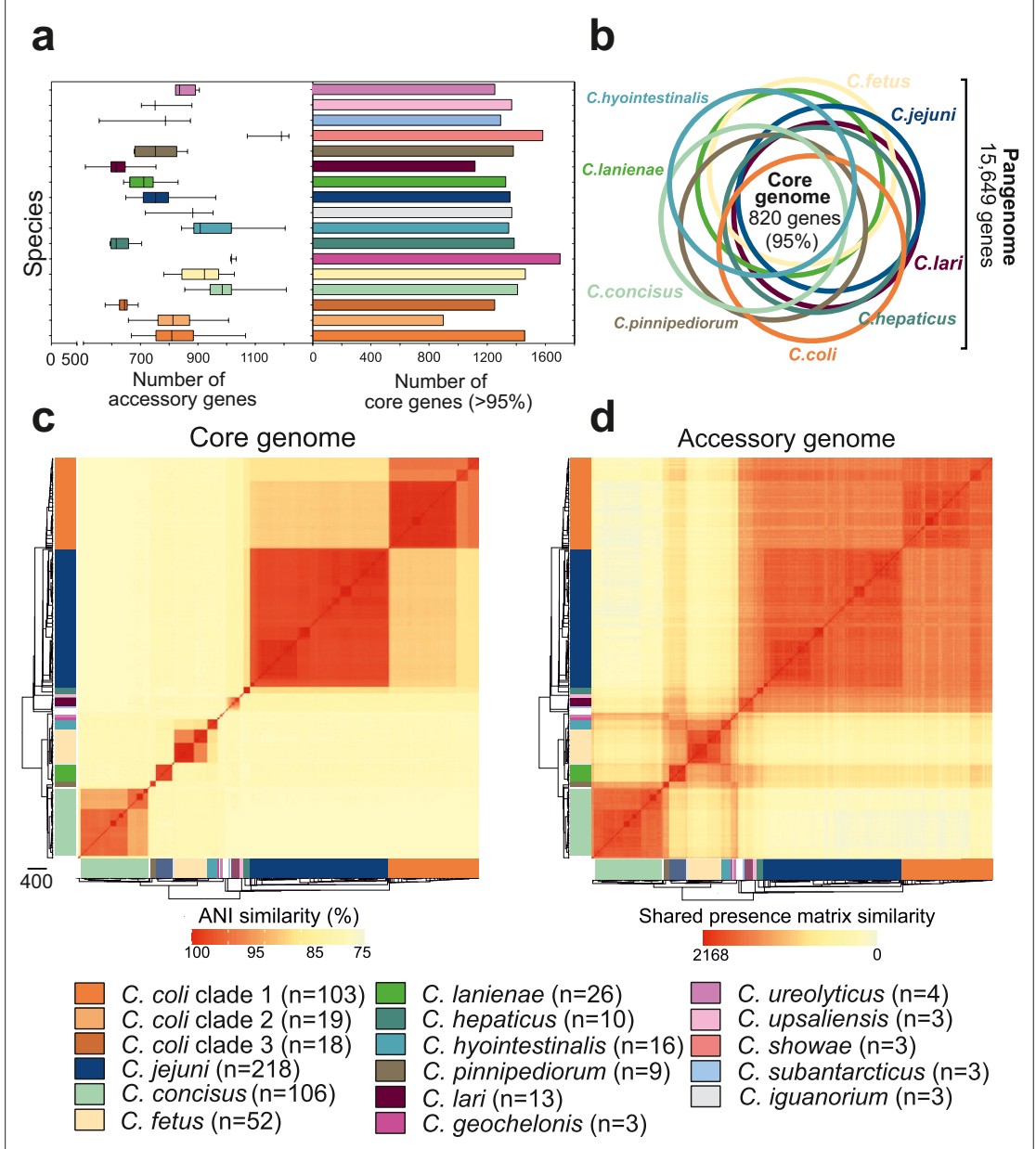

**Figure 3.** Core and accessory genome variation in the genus *Campylobacter*. (**a**) Overall distribution of the total number of accessory genes (left) and core genes (right) per isolate for each *Campylobacter* species (where n ≥ 3 isolates). The number of accessory genes is shown as boxplots (min to max). (**b**) Venn diagram of pangenomes among different *Campylobacter* species (n ≥ 9). The number of core genes shared by all species is illustrated in the center. (**c**) Pairwise average nucleotide identity (ANI) comparison calculated for all 631 *Campylobacter* isolates based upon 820 core genes shared by >95% of isolates. ANI values < 75% are not calculated by FastANI (*Jain et al., 2018*). (**d**) Pairwise accessory genome similarity based upon gene presence or absence at 2,168 non-core loci. The heatmap coloring ranges from yellow (minimum) to red (maximum). The matrices are ordered according to the phylogenetic tree presented in *Figure 2a*. Different colors correspond to *Campylobacter* species with ≥3 isolates.

The online version of this article includes the following source data and figure supplement(s) for figure 3:

**Source data 1.** This file contains the numerical values on which the graphs in *Figure 3* are based.

**Figure supplement 1.** Genome size variation of the *Campylobacter* genus.

**Figure supplement 2.** Gene variation in the genus *Campylobacter*.

**Figure supplement 3.** Accessory gene function in all main *Campylobacter* species.

**Figure supplement 4.** Core genome allelic variation and the effect of recombination.

## Enhanced interspecies recombination among cohabiting species

For *Campylobacter* inhabiting different host species, there is a physical barrier to HGT. However, when there is niche overlap, interspecies recombination can occur, for example, between *C. jejuni* and *C. coli* inhabiting livestock (*Sheppard et al., 2011a*; *Sheppard et al., 2013*; *Sheppard et al., 2008*). To understand the extent to which inhabiting different hosts impedes interspecies gene flow, we quantified recombination among *Campylobacter* species where isolates originated from same host ($x_1$, $y$) and different hosts ($x_2$, $y$) (*Figure 4a*).

ChromoPainterV2 software was used to infer tracts of DNA donated from multiple donor groups, belonging to the same CC but isolated from different hosts to recipient groups (Materials and methods). Among 27 combinations of multiple donor groups and recipient groups, overall, there were more recombining SNPs within hosts than between hosts (*Figure 4b*), and for 10/27 species pairs there was evidence of enhanced within-species recombination ($x_1 \to y > x_2 \to y$; *Figure 4c*). To assess the robustness of the analysis, we included the effect of randomization and repeated the analysis by assigning random hosts for every strain (*Figure 4—figure supplement 1*). In the 10 pair species comparisons where $x_1 \to y > x_2 \to y$, we detected 174,594 within-host recombining SNPs (mapped to 473 genes; 28.8% of NCTC11168 genes) and 109,564 between-host recombining SNPs (mapped to 395 genes; 24.05% of NCTC11168 genes). From the 473 within-host recombining genes, 45 genes contained the highest number (>95th percentile) of recombining SNPs (*Figure 4—figure supplement 2*, *Figure 4—figure supplement 3*, *Supplementary file 2*). These genes have diverse inferred functions including metabolism, cell wall biogenesis, DNA modification, transcription, and translation (*Supplementary file 2*).

Interspecies recombination was observed for isolates sampled from chickens between generalist lineages CC21 and CC45 (donors; *C. jejuni*) and generalist CC828 (recipient; *C. coli*). These lineages appear to have high recombination to mutation (*r/m*) ratio as inferred by ClonalFrameML (*Supplementary file 3*). DNA from generalist *C. jejuni* CC45 was introduced into three *Campylobacter* species, including *C. hepaticus* (chicken), *C. concisus* GSI and GSII (clinical), and *C. ureolyticus* (clinical) (*Figure 4c*, *Figure 4—figure supplement 2*, *Figure 4—figure supplement 3 Supplementary file 4*). CC 45 had the highest *r/m* ratio from all other lineages or species involved in the comparisons (*Supplementary file 3*). There was increased recombination in genomes sampled from cattle between *C. jejuni* CC61 (donor; *C. jejuni*) and *C. fetus* and *C. hyointestinalis* (recipients) with 71.75% of all within-host recombining SNPs from all 10 comparisons detected in these two pairs (*Figure 4c*, *Figure 4—figure supplement 2*, *Figure 4—figure supplement 3*, *Supplementary file 4*). Agricultural-associated *C. jejuni* CC61 and *C. fetus* subsp. *venerealis* involved in these comparisons were among the lineages and subspecies with the highest *r/m* ratios (*Supplementary file 3*). The cattle-associated CC61 has previously been described as highly recombinant and has been associated with rapid clonal expansion and adaptation in cattle (*Mourkas et al., 2020*).

## The within-host mobilome

Bacteria inhabiting the same niche may benefit from functionality conferred by similar gene combinations. Recombination can promote the dissemination of adaptive genetic elements among different bacterial species. Therefore, we postulated that the genes that recombine most among species (>95th percentile) will include those that are potentially beneficial in multiple genetic backgrounds. To investigate this, we quantified mobility within the genome identifying recombining SNPs found in more than one species comparison (*Figure 5a*). These SNPs mapped to 337 genes (20.52% of the NCTC11168 genes; 2.15% of the pangenome) (*Figure 5a*, *Supplementary file 5*). We found that 32 of those genes (9.49%) have also been found on plasmids (*Supplementary file 5*). A total of 16 genes showed elevated within-host interspecies recombination in more than five species pairs (*Figure 5c*, *Supplementary file 5*). Genes included *cmeA* and *cmeB*, which are part of the predominant efflux pump CmeABC system in *Campylobacter*. Sequence variation in the drug-binding pocket of the *cmeB* gene has been linked to increased efflux function leading to resistance to multiple drugs (*Yao et al., 2016*). Many of the same antimicrobial classes are used in human and veterinary medicine, and this may be linked to selection for AMR *Campylobacter,* which are commonly isolated from livestock (*Livermore, 2007*). To investigate this further, we compared the genomes of all 631 isolates in our dataset to 8,762 known antibiotic resistance genes from the Comprehensive Antibiotic Resistance Database (CARD) (*Jia et al., 2017*), ResFinder (*Zankari et al., 2012*), and the National Center for Biotechnology

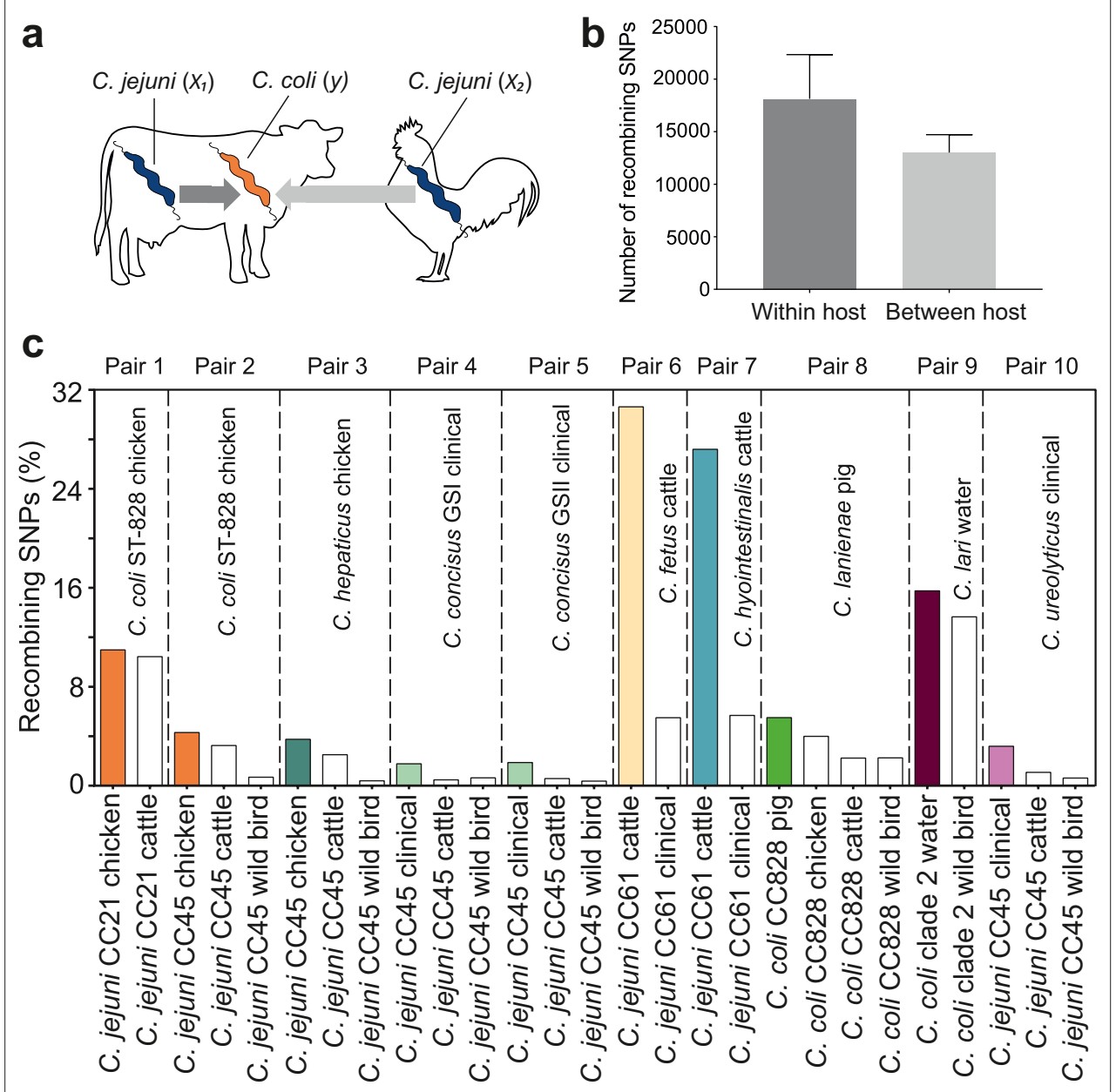

**Figure 4.** Elevated within-host interspecies recombination and donor–recipient comparisons. (**a**) A hypothesis depicting the relationships between *Campylobacter* species, *C. jejuni* ($x_1$, $x_2$) and *C. coli* ($y$), when found in the same or in different hosts. (**b**) Number of recombining SNPs within and between host as inferred by chromosome painting analysis for all donor–recipient species comparisons. The error bar represents the standard error of the mean (SEM). (**c**) The figure shows the number of donated SNPs in 10 donor–recipient pair species comparisons. The proportion (%) of recombining SNPs with >90% probability of copying from a donor to a recipient genome is illustrated on the y-axis. All donor groups are shows in the x-axis. All colored boxes correspond to comparison where donors and recipients are found in the same host.

The online version of this article includes the following figure supplement(s) for figure 4:

**Figure supplement 1.** Probability of the recipient genomes sharing DNA with each donor groups is illustrated as box whiskers (white) for every donor–recipient comparison for all 10 pairs that supported our hypothesis.

**Figure supplement 2.** Genome position of genes containing recombining SNPs.

**Figure supplement 3.** Genes ranked in ascending order of the number of recombining SNPs they contain as inferred by chromosome painting analysis for all 10 species comparisons.

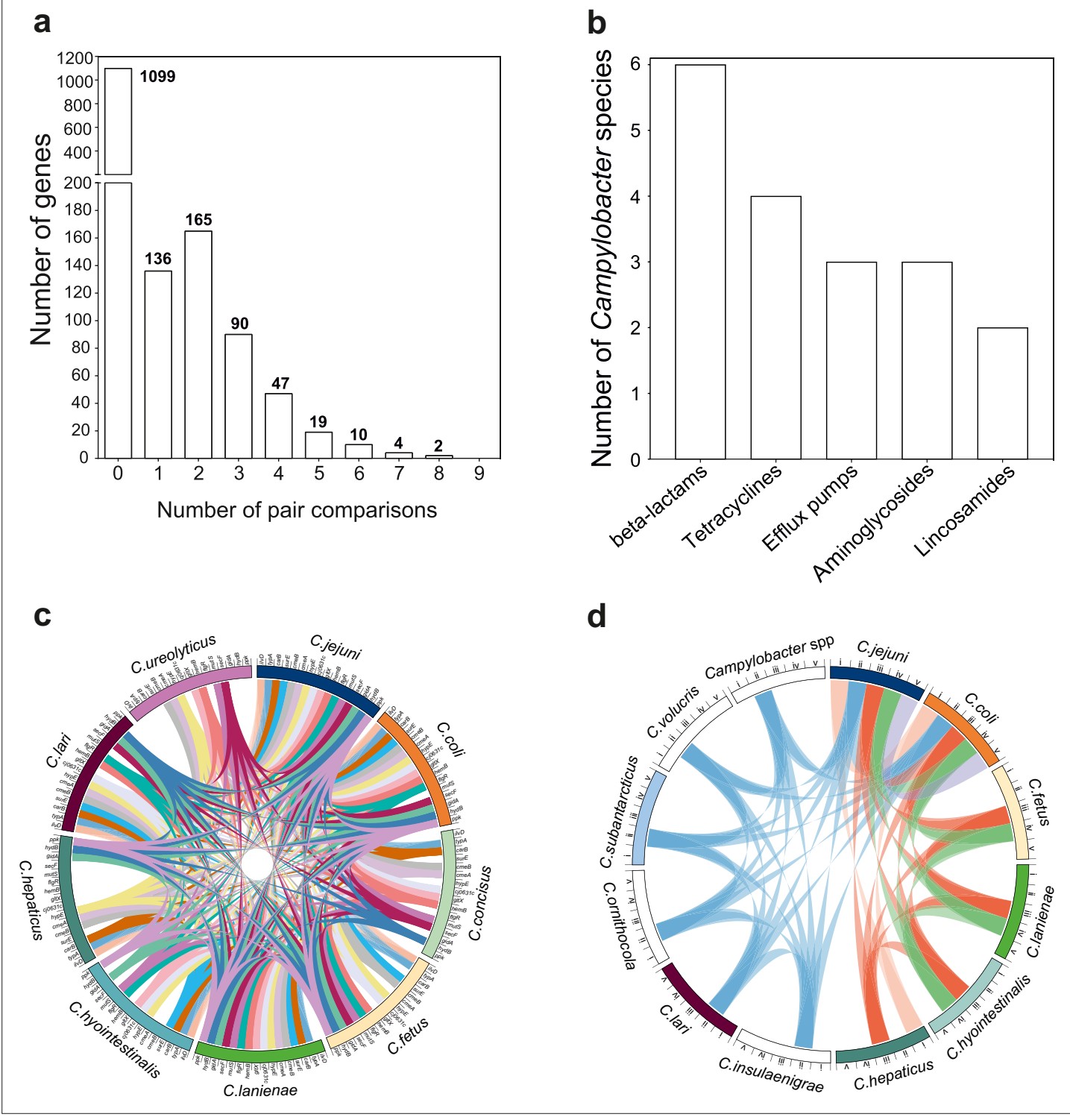

**Figure 5.** The mobilome of the *Campylobacter* genus. (**a**) The graph illustrates the proportion of recombining genes in 10 different species comparisons. The number of species pairs in which the gene was found to recombine is shown on the *x*-axis, and the number of genes in each category is given on the *y*-axis. The exact number of genes found in each group comparison is shown on the top of each box. (**b**) Number of *Campylobacter* species harboring antimicrobial resistance (AMR) genes that belong to efflux pumps and four different antibiotic classes that are shown on the *x*-axis. (**c**) The circos plot indicates the 16 genes involved in recombination in >5 donor–recipient pair species comparisons. Gene matches are indicated by joining lines, colored differently for each gene. Gene names are shown around the perimeter for each *Campylobacter* species. (**d**) The circos plot indicates the sharing of AMR genes associated with efflux pumps and four antibiotic classes among *Campylobacter* species. Presence of at least one

*Figure 5 continued on next page*

*Figure 5 continued*

gene (not necessarily the same gene) conferring resistance to a specific antibiotic class is indicated by joining lines, colored differently for each drug class. Efflux pumps (i), β-lactams (ii), tetracyclines (iii), aminoglycosides (iv), and lincosamides (v) are shown around the perimeter for each *Campylobacter* species.

The online version of this article includes the following source data and figure supplement(s) for figure 5:

**Source data 1.** This file contains the numerical values on which the graphs in *Figure 5b–d* are based.

**Figure supplement 1.** Presence of antimicrobial resistance (AMR) genes in the *Campylobacter* genus.

**Figure supplement 2.** Genetic organization of antimicrobial resistance (AMR) genes in *Campylobacter*.

Information (NCBI) databases. Homology (>75%) was found for 42 AMR determinants associated with multidrug efflux pumps, aminoglycosides, tetracyclines, and β-lactams (*Figure 5b*, *Figure 5—figure supplement 1*, *Figure 5—source data 1*). Species that contained >40% isolates from livestock, including *C. jejuni*, *C. coli*, *C. lanienae*, *C. hepaticus*, *C. hyointestinalis,* and *C. fetus,* contained far more AMR determinants (*Figure 5d*, *Figure 5—figure supplement 1*, *Figure 5—source data 1*). AMR genes are often collocated in the genome (*Mourkas et al., 2019*), and our analysis revealed several gene clusters (*Figure 5—figure supplement 2*) that have been described in previous studies (*Mourkas et al., 2019*; *Abril et al., 2010*). These findings are consistent with HGT-mediated circulation of AMR genes among different *Campylobacter* species and support the hypotheses that ecology drives gene pool transmission (*Sheppard et al., 2018*; *Mourkas et al., 2019*).

*Campylobacter* host transmission and virulence have been linked with biofilm formation and changes in surface polysaccharides (*Szymanski et al., 2003*; *McLennan et al., 2008*). The *carB* gene showed elevated within-host interspecies recombination in eight species pair comparisons (*Figure 5c*, *Supplementary file 5*). This gene encodes a carbamoylphosphate synthase that has been associated with biosynthesis of substrates for many polysaccharides and is known to contain transposon insertion sites upstream of its genomic position (*McLennan et al., 2008*). Other genes with elevated within-host interspecies transfer (>7 species pairs) included *typA* (*Figure 5c*, *Supplementary file 5*), a translator regulator for GTPase and *gltX* (*Figure 5c*), *a glutamate-tRNA ligase*, promoting survival under stress conditions (*Margus et al., 2007*; *Semanjski et al., 2018*). Other genes included *gidA* and *hydB* associated with virulence (*Mikheil et al., 2012*) and hydrogenase enzyme activity (respiratory pathway in *C. concisus*, *Benoit and Maier, 2018*), respectively. By considering genes that overcome barriers to interspecies recombination and establish in multiple new genetic backgrounds, it may be possible to infer important phenotypes that allow bacteria to adapt to different hosts and environments.

## Discussion

Phylogenetic reconstruction of the genus *Campylobacter* revealed a highly structured population. Distinct core genome clustering largely supported known classification for species, subspecies (*C. fetus*, *Iraola et al., 2017*), genomospecies (*C. concisus*, *Kirk et al., 2018*), and clades (*C. coli Sheppard et al., 2008*). Also consistent with previous studies, certain species are principally associated with a specific host niche. For example, *C. fetus* subsp. *testudinum*, *C. iguanorium,* and *C. geochelonis* were only sampled from reptile species, and *C. pinnipediorum* was only sampled from seals. However, for several species there was clear evidence for host generalism, including *C. jejuni*, *C. coli*, and *C. lari*, all of which were sampled from multiple hosts (*Griekspoor et al., 2013*; *Cody et al., 2015*). It is clear that the hosts with the greatest diversity of *Campylobacter* species were agricultural animals (and humans) (*Figure 2—figure supplement 3*). While this undoubtedly reflects oversampling of these sources to some extent, the cohabitation of species in the same host niche potentially provides opportunities for interspecies HGT.

Initial evidence of interspecies gene flow came from comparison of ANI and the accessory genome gene presence/absence for all isolates. In each case, patterns of genetic similarity largely mirrored the phylogeny. However, consistent with previous studies (*Sheppard et al., 2013*), there was clear evidence of elevated homologous and non-homologous recombination between some species. For example, core genome ANI was higher between *C. jejuni* and *C. coli* clade 1 compared to other *C. coli* clades (*Figure 3—source data 1*). The evidence for non-homologous gene sharing was even more

striking with accessory genome sharing across considerable genetic distances (*Figure 3—source data 1*), exemplified by *C. lanienae,* which shares accessory genes with most other *Campylobacter* species.

To quantify the extent to which ecological barriers influenced interspecies gene flow, it was necessary to focus on donor–recipient species pairs where there was evidence of elevated HGT in the same (sympatry) compared to different (allopatry) hosts. Perhaps unsurprisingly, this was not the case for all species comparisons. Interacting factors could lead to genetic isolation even when species inhabit the same host. First, rather than being a single niche, the host represents a collection of subniches with varying degrees of differentiation. For example, gut-associated bacteria in the same intestinal tract have been shown to occupy different microniches (*Hayashi et al., 2005*) and more striking segregation may be expected between *C. hepaticus* inhabiting the liver in poultry (*Van et al., 2016*) and gut-dwelling *C. jejuni* and *C. coli* in the same host. Second, there is potential for the resident microbiota to influence the colonization potential of different *Campylobacter* species and therefore the opportunity for genetic exchange, for example, through succession (*Lu et al., 2003*) and inhibition of transient species by residents, as seen in some other bacteria in humans (*Stecher et al., 2010*; *van Elsas et al., 2012*; *Nowrouzian et al., 2005*).

Continued exposition of the microecology of subniches is important, but for 10 species comparisons, there was clear evidence of enhanced within-host gene flow allowing quantitative analysis of ecological barriers to gene flow. Specifically, there was on average a three-fold increase in recombination among species pairs inhabiting the same host. In some cases, this was greater, with 5–6 times more recombination among cohabiting species *C. jejuni* and *C. hyointestinalis/C. fetus* in cattle. In absolute terms, this equates to approximately 30% of all recorded SNPs in the recipient species being the result of introgression. To place this in context, if greater than half (51%) of the recorded SNPs resulted from interspecies recombination then the forces of species convergence would be greater than those that maintain distinct species. If maintained over time, these relative rates could lead to progressive genetic convergence unless countered by strong genome-wide natural selection against introgressed DNA.

Quantitative SNP-based comparisons clearly ignore one very important factor. Specifically, that recombined genes that do not reduce the fitness of the recipient genome (provide an adaptive advantage) will remain in the population while others will be purged through natural selection. Therefore, by identifying genomic hotspots of recombination and the putative function of genes that recombine between species, it is possible to understand more about microniche segregation and the host-adapted gene pool. Of the 35 genes with evidence of enhanced within-host HGT in ≥5 species pairs, several were linked to functions associated with proliferation in, and exploitation of, the host. For example, the *carB* gene, encoding the large subunit of carbamoylphosphatase associated with polysaccharide biosynthesis, recombined in eight cohabiting species pairs and is potentially linked to enhanced virulence and growth (*McLennan et al., 2008*). In addition, other highly mobile genes, including *typA* and *gltX,* are associated with survival and proliferation in stress conditions (*Margus et al., 2007*; *Semanjski et al., 2018*), and *hydB* is linked to NiFe hydrogenase and nickel uptake that is essential for the survival of *C. jejuni* in the gut of birds and mammals (*Howlett et al., 2012*).

Some genes showed evidence of elevated recombination in a specific host species. For example, the *glmS* and *napA* genes in cohabiting *Campylobacter* species in cattle. In many bacteria, analogs of *glmS* have multiple downstream integration-specific sites (Tn7) (*Choi and Kim, 2009*), which may explain the mobility of this gene. Explaining the mobility of *napA* is less straightforward, but this gene is known to encode a nitrate reductase in *Campylobacter* (*Pittman et al., 2007*) in microaerobic conditions, which may be ecologically significant as the accumulation of nitrate in slurry, straw, and drainage water can be potentially toxic to livestock mammals (*Alexander et al., 2009*).

Factors such as host physiology, diet, and metabolism undoubtedly impose selection pressures upon resident bacteria, and the horizontal acquisition of genes provides a possible vehicle for adaptation. However, the widespread use of antimicrobials by humans, pets, and livestock production (*Teuber, 2001*; *Price et al., 2015*) provides another major ecological barrier to niche colonization. We found that *gyrA* was among the most recombinogenic genes in *Campylobacter* in chickens. This is important as a single mutation in this gene is known to confer resistance to ciprofloxacin (*Luo et al., 2003*). While the rising trend in fluoroquinolones resistance in *Campylobacter* from humans and livestock (*Sproston et al., 2018*) may result from spontaneous independent mutations, it is likely that it is accelerated by HGT. However, there is currently no clear evidence for the transfer of resistant versions of *gyrA*.

Interspecies recombination of AMR genes has been observed between *C. jejuni* and *C. coli* isolates from multiple sources including livestock, human, and sewage (**Mourkas et al., 2019**). Consistent with this, we found AMR genes present in strains from 12 *Campylobacter* species in multiple hosts (**Figure 5— figure supplement 2**). In some cases, strains from phylogenetically closely related species (*C. fetus* and *C. hyointestinalis*) isolated from cattle shared the same AMR gene cluster (*tet44* and *ant(6)-Ib*) described before in *C. fetus* subsp. *fetus* (**Abril et al., 2010**), indicating the circulation of colocalized AMR genes among related species and host niche gene pools. Strikingly, the efflux pump genes *cmeA* and *cmeB*, associated with multidrug resistance (MDR), were highly mobile among *Campylobacter* species with evidence of elevated within-host interspecies recombination in >7 species pairs. Furthermore, the *gltX* gene, which when phosphorylated by protein kinases promotes MDR (**Semanjski et al., 2018**), was also among the most introgressed genes. While a deeper understanding of gene interactions, epistasis, and epigenetics would be needed to prove that the lateral acquisition of AMR genes promotes niche adaptation, these data do suggest that HGT may facilitate colonization of antimicrobial-rich host environments, potentially favoring the spread of genes into multiple genetic backgrounds.

In conclusion, we show that species within the genus *Campylobacter* include those that are host restricted as well as host generalists. When species cohabit in the same host, ecological barriers to recombination can be perforated, leading to considerable introgression between species. While the magnitude of introgression varies, potentially reflecting microniche structure within the host, there is clear evidence that ecology is important in maintaining genetically distinct species. This parallels evolution in some interbreeding eukaryotes, such as Darwin's Finches, where fluctuating environmental conditions can change the selection pressures acting on species inhabiting distinct niches, potentially favoring hybrids (**Mallet, 2007**; **Grant and Grant, 1992**). Consistent with this, the host landscape is changing for *Campylobacter,* with intensively reared livestock now constituting 60–70% of bird and mammal biomass on earth, respectively (**Bar-On et al., 2018**). This creates opportunities for species to be brought together in new adaptive landscapes and for genes to be tested in multiple genetic backgrounds. By understanding the ecology of niche segregation and the genetics of bacterial adaptation, we can hope to improve strategies and interventions to reduce the risk of zoonotic transmission and the spread of problematic genes among species.

## Materials and methods
### Isolate genomes

A total of 631 *Campylobacter*, 17 *Arcobacter*, 7 *Sulfurospirillum,* and 5 *Helicobacter* genomes were assembled from previously published datasets (**Supplementary file 1**). Isolates were sampled from clinical cases of campylobacteriosis and feces of chickens, ruminants, wild birds, wild mammals, pets, and environmental sources. Genomes and related metadata were uploaded and archived in the BIGS database (**Sheppard et al., 2012**). Quality control was performed based on the genome size, number of contigs, and N50 and N95 contig length using the integrated tools in BIGS database. All assembled contigs were further screened for contamination and completeness using CheckM (**Parks et al., 2015**; **Supplementary file 1**). All assembled genomes can be downloaded from FigShare (doi: 10.6084/m9.figshare.15061017). Comparative genomics analyses focused on the *Campylobacter* genomes representing 30 species including *C. avium* (n = 1); *C. coli* (n = 143); *C. concisus* (n = 106); *C. corcagiensis* (n = 1); *C. cuniculorum* (n = 2); *C. curvus* (n = 2); *C. fetus* (n = 52); *C. geochelonis* (n = 3); *C. gracilis* (n = 2); *C. helveticus* (n = 1); *C. hepaticus* (n = 10); *C. hominis* (n = 1); *C. hyointestinalis* (n = 16); *C. iguanorium* (n = 3); *C. insulaenigrae* (n = 1); *C. jejuni* (n = 218); *C. lanienae* (n = 26); *C. lari* (n = 13); *C. mucosalis* (n = 1); *C. ornithocola* (n = 1); *C. peloridis* (n = 1); *C. pinnipediorum* (n = 9); *C. rectus* (n = 1); *C. showae* (n = 3); *C. sputorum* (n = 1); *C. subantarcticus* (n = 3); *C. upsaliensis* (n = 3); *C. ureolyticus* (n = 4); *C. volucris* (n = 2); and *Campylobacter* sp. (n = 1) (**Supplementary file 1**). Genomes belonging to *C. jejuni* and *C. coli* species were selected to represent a wide range of hosts, sequence types, and CCs and reflect the known population structure for these two species. For other *Campylobacter* species, all genomes that were publicly available at the time of this study were included in the analysis (**Supplementary file 1**).

### Pangenome characterization and phylogenetic analysis

Sequence data were analyzed using PIRATE, a fast and scalable pangenomics tool that allows for ortholog gene clustering in divergent bacterial species (**Bayliss et al., 2019**). Genomes were

annotated in Prokka (*Seemann, 2014*) using a genus database comprising well-annotated *C. jejuni* strains NCTC11168, 81116, 81-176 and M1, and plasmids pTet and pVir in addition to the already existing databases used by Prokka (*Seemann, 2014*). Briefly, annotated genomes were used as input for PIRATE. Nonredundant representative sequences were produced using CD-HIT, and the longest sequence was used as a reference for sequence similarity interrogation using BLAST/DIAMOND. Gene orthologs were defined as 'gene families' and were clustered in different MCL thresholds, from 10 to 98% sequence identity (10, 20, 30, 40, 50, 60, 70, 80, 90, 95, 98). Higher MCL thresholds were used to identify allelic variation within different loci. An inflation value of 4 was used to increase the granularity of MCL clustering between gene families. BLAST high-scoring pairs with a reciprocal minimum length of 90% of the query/subject sequence were excluded from MCL clustering to reduce the number of spurious associations between distantly related or conserved genes (*Sahl et al., 2014*). This information was used to generate gene presence/absence and allelic variation matrices. A core gene-by-gene multiple sequence alignment (*Sheppard et al., 2012*) was generated using MAFFT (*Katoh et al., 2002*) comprising genes shared >95% of isolates. Phylogenetic trees, based on core gene-by-gene alignments, were reconstructed using the maximum-likelihood algorithm implemented in RAxML v8.2.11 (*Stamatakis, 2014*) with GTRGAMMA as substitution model.

## Quantifying core and accessory genome variation

The degree of genetic differentiation between species was investigated gene-by-gene as in previous studies (*Sheppard et al., 2013*; *Didelot et al., 2007*) by calculating the ANI of all 631 *Campylobacter* genomes using FastANI v.1.0 (*Jain et al., 2018*). The analysis generated a lower triangular matrix with the lowest ANI value at 75% (as computed by FastANI). A comparable gene presence/absence matrix was produced using PIRATE and was further used to generate a heatmap of accessory genome similarity based upon gene presence or absence. Subsequently, all *Campylobacter* genomes were screened for the presence of AMR genes against the CARD (*Jia et al., 2017*), ResFinder (*Zankari et al., 2012*), and NCBI databases. All *Campylobacter* genomes were further screened for the presence of phage, conjugative elements, and plasmid DNA using publicly available online databases to investigate the effect of other transfer mechanisms. First, we used the PHAge Search Tool Enhanced Release (PHASTER) (*Arndt et al., 2016*) to identify and annotate prophage sequences within our genomes. A total of 86% (254/297) of the genomes used in chromosome painting were found to have DNA sequence of phage origin. Second, we used Iceberg 2.0 (*Liu et al., 2019*) for the detection of integrative and conjugative elements, identifying 32 ICEs in 19% (56/297) of the genomes used in the chromosome painting analysis. Finally, we used MOB-suite software for clustering, reconstructing, and typing of plasmids from draft assemblies (*Robertson and Nash, 2018*; *Robertson et al., 2020*). A positive hit was defined when a gene had >75% nucleotide identity over >50% of the sequence length showing that 32 genes identified in the recombination analysis have also been located on plasmids. A gene presence/absence matrix for every AMR gene was generated for every genome. Genomes carrying AMR genes were screened to characterize the location of adjacent genes using SnapGene software (GSL Biotech; available at https://www.snapgene.com/), as previously described (*Mourkas et al., 2019*). The number of core SNPs was identified using SNP-sites (v2.3.2) (*Page et al., 2016*).

## Inference of recombination

Each combination of a recipient group and multiple donor groups (belonging to the same CC but isolated from different hosts) was selected to compare the extent of interspecies recombination into the recipient genomes. Each donor group consisted of eight isolates to avoid the influence of difference in sample size on estimation of the extent of interspecies recombination. Each recipient group included at least four isolates. We excluded *C. jejuni* and *C. coli* clade 1 genomes isolated from seals and water as these most likely represent spillover events and not true host-segregated populations. Briefly, we conducted a pairwise genome alignment between reference genome NCTC11168 and one of the strains included in the donor–recipient analysis using progressiveMauve (*Darling et al., 2010*). This enabled the construction of positional homology alignments for all genomes regardless gene content and genome rearrangements, which were then combined into a multiple whole-genome alignment, as previously described (*Yahara et al., 2018*). ChromoPainterV2 software was used to calculate the amount of DNA sequence that is donated from a donor to a recipient group (*Lawson et al., 2012*). Briefly, for each donor–recipient pair, SNPs in which >90% recipient individuals had

recombined with the donor group were considered in the analysis. These SNPs were mapped to genomic regions and specific genes were identified. A total of 258,444 (96.83%) recombining SNPs mapped to 558 genes of the NCTC11168 reference strain with >90% probability of copying from a donor to a recipient strain. Genes containing the highest number of recombining SNPs were considered for subsequent analyses (>95th percentile) (*Supplementary file 2*). ClonalFrameML (*Didelot and Wilson, 2015*) was used to infer the relative number of substitutions introduced by recombination ($r$) and mutation ($m$) as the ratio $r/m$ as previously described (*Mourkas et al., 2020*).

## Acknowledgements

This work was supported by Wellcome Trust grants 088786/C/09/Z and Medical Research Council (MRC) grants MR/M501608/1 and MR/L015080/1 awarded to SKS. The computational calculations were performed at the Human Genome Center at the Institute of Medical Science (University of Tokyo) and the National Institute of Genetics.

## Additional information

### Funding

| Funder | Grant reference number | Author |
|---|---|---|
| Medical Research Council | MR/M501608/1 | Samuel K Sheppard |
| Medical Research Council | MR/L015080/1 | Samuel K Sheppard |
| Wellcome Trust | 088786/C/09/Z | Samuel K Sheppard |

The funders had no role in study design, data collection and interpretation, or the decision to submit the work for publication.

### Author contributions

Evangelos Mourkas, Conceptualization, Data curation, Formal analysis, Investigation, Methodology, Software, Validation, Visualization, Writing - original draft, Writing – review and editing; Koji Yahara, Data curation, Formal analysis, Methodology, Software, Validation, Visualization, Writing – review and editing; Sion C Bayliss, Data curation, Formal analysis, Methodology, Software, Writing – review and editing; Jessica K Calland, Håkan Johansson, Leonardos Mageiros, Zilia Y Muñoz-Ramirez, Santiago Sandoval-Motta, Javier Torres, Data curation, Formal analysis; Grant Futcher, Data curation; Guillaume Méric, Conceptualization; Matthew D Hitchings, Keith A Jolley, Martin CJ Maiden, Jonas Waldenström, Data curation, Resources; Patrik Ellström, Resources; Ben Pascoe, Conceptualization, Data curation, Formal analysis, Supervision, Validation, Writing – review and editing; Samuel K Sheppard, Conceptualization, Funding acquisition, Investigation, Project administration, Supervision, Writing – review and editing

### Author ORCIDs

Evangelos Mourkas http://orcid.org/0000-0002-7411-4743
Koji Yahara http://orcid.org/0000-0003-4289-1115
Sion C Bayliss http://orcid.org/0000-0002-5997-2002
Leonardos Mageiros http://orcid.org/0000-0002-0846-522X
Zilia Y Muñoz-Ramirez http://orcid.org/0000-0001-8673-0191
Guillaume Méric http://orcid.org/0000-0001-6288-9958
Javier Torres http://orcid.org/0000-0003-3945-4221
Keith A Jolley http://orcid.org/0000-0002-0751-0287
Martin CJ Maiden http://orcid.org/0000-0001-6321-5138
Jonas Waldenström http://orcid.org/0000-0002-1152-4235
Ben Pascoe http://orcid.org/0000-0001-6376-5121
Samuel K Sheppard http://orcid.org/0000-0001-6901-3203

### Decision letter and Author response

Decision letter https://doi.org/10.7554/eLife.73552.sa1
Author response https://doi.org/10.7554/eLife.73552.sa2

## Additional files

### Supplementary files
- Supplementary file 1. Isolate information about the genomes used in this study.
- Supplementary file 2. Within-host highly (>95th percentile) recombining genes.
- Supplementary file 3. Recombination parameters as calculated by ClonalFrameML.
- Supplementary file 4. Quantifying recombination between cohabiting species using ChromoPainter.
- Supplementary file 5. Genes involved in interspecies recombination in 10 species comparisons.
- Transparent reporting form

### Data availability
Genomes sequenced as part of other studies are archived on the Short Read Archive associated with BioProject accessions: PRJNA176480, PRJNA177352, PRJNA342755, PRJNA345429, PRJNA312235, PRJNA415188, PRJNA524300, PRJNA528879, PRJNA529798, PRJNA575343, PRJNA524315 and PRJNA689604. Additional genomes were also downloaded from NCBI and pubMLST (http://pubmlst.org/campylobacter). Contiguous assemblies of all genome sequences compared are available at the public data repository Figshare (doi: 10.6084/m9.figshare.15061017) and individual project and accession numbers can be found in Supplementary file 1.

The following dataset was generated:

| Author(s) | Year | Dataset title | Dataset URL | Database and Identifier |
|---|---|---|---|---|
| Pascoe B | 2021 | The ecology of interspecies recombination among the zoonotic bacterium Campylobacter | https://doi.org/10.6084/m9.figshare.15061017 | figshare, 10.6084/m9.figshare.15061017 |

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
