## [Editor Report]

This article will be of broad interest to readers who work in bacterial genomics, particularly those conducting research on *Campylobacter* genomics. This article substantially advances the field by quantifying horizontal gene transfer among sympatric and allopatric species in natural populations, and demonstrating enhanced horizontal gene transfer between *Campylobacter* species that colonize the same host species.

---

## [Decision Letter]

**Decision letter after peer review:**

Thank you for submitting your article "Host ecology regulates interspecies recombination in bacteria of the genus *Campylobacter*" for consideration by *eLife*. Your article has been reviewed by 2 peer reviewers, and the evaluation has been overseen by a Reviewing Editor and Gisela Storz as the Senior Editor. The reviewers have opted to remain anonymous.

Essential revisions:

1. A clearer explanation of the cutoffs/stats/etc used for determining the HGT regions is needed.

2. Can the authors provide information on whether the accessory HGT genes are located on specific genomic regions?

3. Can the authors comment on evidence of DNA transfer without very gene specific selection pressure in other Campylobacter species as well (similar to that shown by the authors for C. jejuni and C. coli)?

4. Could the authors either provide clear evidence of transfer of resistant versions of gyrA or make it clear that there is no evidence yet.

5. Is there some way of showing which species is more often a donor and which is more often recipient, as in this paper: https://www.nature.com/articles/ng.2895?

6. More specific details about how the strains were selected for the study are needed.

7. All the genomes were from a public database from previously published studies, however there is no mention about how the quality of each genome was checked prior to analysis. Did the genomes get run through CheckM or similar software for screening for contamination and/or completeness? This is a critical analysis for this study.

8. An explanation of how differences in geographical location were taken into consideration during the analysis is needed.

9. There are issues with the referencing of figures throughout the manuscript which need to be addressed (see Reviewer #2 comments).

10. Could the authors improve the colour selection for the different Campylobacter species to make them easier to differentiate, particularly in Figure 3.

11. Line 151 – 153 – "10/30 species isolated from more than one host species". Does this refer to agricultural host species or general host species?

*Reviewer #1:*

I think the paper is well written and shows what everyone suspects, namely plenty of transfer of DNA between various Campylobacter species dependent on where they are co-habiting.

1. I recommend a clearer explanation of the cutoffs/stats/etc used for determining the HGT regions.

2. Are the accessory HGT genes located on specific genomic regions? e.g. phages? conjugative elements? plasmids? I think that should be investigated, as these generally have very different transfer mechanisms. Is it uptake of naked DNA or phage transfection for instance and are we looking at bacteriophages that have jumped hosts which could suggest that very different mechanisms and evolutionary reasons are at play.

3. The conclusion of HGT of very specific DNA regions associated with a niche is not entirely compatible with the conclusion of large scale introgression, resulting in blurring of species boundaries (line 327). Only very small parts of the genome are transferred if its about host adaptation genes. Is there evidence that there is a lot of transfer of DNA without very gene specific selection pressure in other Campylobacter species as well (like what was shown before by the authors for C. jejuni and C. coli)?

4. Transfer of fq resistance providing gyrase is suspected ("highly recombinogenic"), but no real evidence is given (is a resistant version of gyrA of species A found in species B?). Either provide clear evidence of transfer of resistant versions of gyrA or make it clear that there is no evidence yet.

5. Is there some way of showing which species is more often donor and which is more often recipient? Like in this paper: https://www.nature.com/articles/ng.2895.

*Reviewer #2:*

Overall, the manuscript was very interesting and informative, and the results move the world of Campylobacter genomics forward. While some of the results in the manuscript were known at least antidotally this manuscript provides the evidence to confirm these facts, while also presenting additional novel results about Campylobacter recombination.

1. There are major issues with the referencing of figures throughout the manuscript, for example:

a. Line 137 – Figure 1 is referenced but it would make more sense to be Figure 2a, similarly Figure 1b is referenced when there is no Figure 1b.

b. Figure 3b is never referenced in the manuscript.

c. Figures 5c – e is also never referenced in the manuscript.

2. The colors selected for the different Campylobacter species can be difficult to differentiate particularly in Figure 3. I would recommend changing some of the colors, particularly C. lanienae and C. lari are extremely hard to tell differences in Figure 3a.

3. Line 151 – 153 – "10/30 species isolated from more than one host species" – this is unclear if it is agricultural host species or general host species?

4. Line 354 – "humans, and in pets livestock production" should be changed "pets and livestock production"

5. Line 358 – "fluorophinolone" changed to "fluoroquinolones"

6. Line 365 – "subsp. Fetus" changed to "subsp. fetus"

7. Line 387 – "genes to be tested multiple genetic backgrounds" changed to "genes to be tested from multiple backgrounds.

---

## [Author Response]

Reviewer #1:I think the paper is well written and shows what everyone suspects, namely plenty of transfer of DNA between various Campylobacter species dependent on where they are co-habiting.

We thank reviewer 1 for their positive comments. We have addressed all comments in detail below with reference to amendments (line numbers) in the revised submission.

1. I recommend a clearer explanation of the cutoffs/stats/etc used for determining the HGT regions.

We welcome this suggestion to clarify the methods. The horizontally transferred regions were inferred by ChromoPainterV2 as previously described (Lawson et al., 2012). Briefly, for each donor-recipient pair, SNPs in which >90% recipient individuals had recombined with the donor group were considered in the analysis. These SNPs were mapped to genomic regions and specific genes were identified. A total of 258,444 (96.83%) recombining SNPs mapped to 558 genes of the NCTC11168 reference strain with >90% probability of copying from a donor to a recipient strain. Genes containing the highest number of recombining SNPs were considered for subsequent analyses (>95^th^ percentile) (Supplementary File 2). These details have been added to the methods section in the revised submission (lines 484-490).

2. Are the accessory HGT genes located on specific genomic regions? e.g. phages? conjugative elements? plasmids? I think that should be investigated, as these generally have very different transfer mechanisms. Is it uptake of naked DNA or phage transfection for instance and are we looking at bacteriophages that have jumped hosts which could suggest that very different mechanisms and evolutionary reasons are at play.

Consistent with the reviewer’s suggestion, we have screened all of the genomes for the presence of phage, conjugative elements and plasmid DNA using publicly available online databases to investigate the effect of other transfer mechanisms. First, we used the PHAge Search Tool Enhanced Release (PHASTER) (Arndt et al., 2016) to identify and annotate prophage sequences within our genomes. A total of 86% (254/297) of the genomes used in chromosome painting were found to have DNA sequence of phage origin. Second, we used MOB-suite software (Robertson and Nash, 2018) for clustering, reconstruction and typing of plasmids from draft assemblies. This showed that 32 genes identified in the recombination analysis have also been located on plasmids. Third, we used Iceberg 2.0 (Liu et al., 2019) for the detection of integrative and conjugative elements. We identified 32 ICEs in 18.85% (56/297) of the genomes used in the chromosome painting analysis. These findings have been added in the revised submission (lines 455-465).

3. The conclusion of HGT of very specific DNA regions associated with a niche is not entirely compatible with the conclusion of large scale introgression, resulting in blurring of species boundaries (line 327). Only very small parts of the genome are transferred if its about host adaptation genes. Is there evidence that there is a lot of transfer of DNA without very gene specific selection pressure in other Campylobacter species as well (like what was shown before by the authors for C. jejuni and C. coli)?

We agree that there is a difference between HGT of adaptive genes and large-scale genome-wide introgression. Discussion in this paragraph (lines 327-337) is included to illustrate the extent to which host barriers may maintain distinct species. Specifically, from a purely numerical point of view, if SNPs resulting from interspecies gene flow (HGT) exceed those from mutation then the forces of convergence are greater than the forces that maintain divergent species. This is obviously an oversimplification because it ignores selection that will favour certain genetic changes and purge others. One of the challenges when estimating the magnitude of HGT between species is that only imported DNA that is not detrimental to the recipient genome will be observed in the population (that we sample). Hence, in some cases, much of the imported DNA will be purged leaving behind only small fragments of larger HGT events. While we consider it useful to place our findings in the context of forces that maintain species, more isolate genomes would be needed to conduct large-scale quantitative analysis comparable to that which concluded species convergence between C. jejuni and C. coli.

4. Transfer of fq resistance providing gyrase is suspected ("highly recombinogenic"), but no real evidence is given (is a resistant version of gyrA of species A found in species B?). Either provide clear evidence of transfer of resistant versions of gyrA or make it clear that there is no evidence yet.

As fluoroquinolone resistance results from a single SNP in the multi-copy gyrA gene, and this occurs multiple times in divergent lineages with different gyrA alleles, it is very difficult to quantify HGT at this locus. Consistent with the reviewer’s comment we have made clear that there is currently no evidence of transfer of resistant versions of gyrA in the revised submission (line 369-370).

5. Is there some way of showing which species is more often donor and which is more often recipient? Like in this paper: https://www.nature.com/articles/ng.2895.

We are familiar with this excellent paper and the paper of Croucher et al., (Science, 2011) in which the r/m methodology is described in more detail. In a very large isolate genome collection, such as the Maela S. pneumoniae dataset, this method is appropriate. However, where there are lower numbers of isolate genomes, the principal confounding effect of this type of analysis becomes more pronounced. Specifically, it may be very difficult to differentiate donors from recipients. Especially when the number of individuals in the donor and recipient population are different. For example, if population X contains 100 genomes and population Y contains 20 genomes, then the direction of gene flow may more likely be inferred from X to Y. While this can be overcome in the experimental design, we have preferred a different approach. By defining inferred donor and recipient populations in the Chromopainter input, broadly matching donor numbers, we can address our specific hypothesis that gene flow is greater from X1 to Y than from Χ2 to Y (Figure 4a). That is to say that recombination is greater within host than between hosts.

Reviewer #2:Overall, the manuscript was very interesting and informative, and the results move the world of Campylobacter genomics forward. While some of the results in the manuscript were known at least antidotally this manuscript provides the evidence to confirm these facts, while also presenting additional novel results about Campylobacter recombination.

We thank reviewer 2 for their positive comments. We have addressed all comments in detail below with reference to amendments (line numbers) in the revised submission.

1. There are major issues with the referencing of figures throughout the manuscript, for example:a. Line 137 – Figure 1 is referenced but it would make more sense to be Figure 2a, similarly Figure 1b is referenced when there is no Figure 1b.b. Figure 3b is never referenced in the manuscript.c. Figures 5c – e is also never referenced in the manuscript.

We thank the reviewer for these points. We have revised the resubmission to make sure that all figures have been correctly referenced throughout the text. Specifically:

a. Figure 1 has been replaced with figure 2a in line 137. Figure 1b has been replaced with figure 2b in lines 148, 149 and 157.

b. Figure 1b has been replaced with figure 3b in lines 166 and 167. Additionally, Figure 2a has been replaced with Figure 3a in line 170 and figure 3c with figure 4c in line 240.

c. Figure 5b is now referenced in lines 266, figure 5c in lines 256, 277, 281, and figure 5d in line 269.

2. The colors selected for the different Campylobacter species can be difficult to differentiate particularly in Figure 3. I would recommend changing some of the colors, particularly C. lanienae and C. lari are extremely hard to tell differences in Figure 3a.

As suggested, we have now changed the colours for C. lanienae and C. lari in all main and supplementary figures.

3. Line 151 – 153 – "10/30 species isolated from more than one host species" – this is unclear if it is agricultural host species or general host species?

This has now been corrected and reads: “…with 10/30 Campylobacter species isolated from more than one source…” in line 159.

4. Line 354 – "humans, and in pets livestock production" should be changed "pets and livestock production"

“humans, and in pets livestock production” has been replaced with “pets and livestock production” (lines 363-364).

5. Line 358 – "fluorophinolone" changed to "fluoroquinolones"

"fluorophinolone" has been replaced with "fluoroquinolones" (line 367).

6. Line 365 – "subsp. Fetus" changed to "subsp. fetus"

‘C. fetus subsp. Fetus’ has been replaced with ‘C. fetus subsp. fetus’ (line 375).

7. Line 387 – "genes to be tested multiple genetic backgrounds" changed to "genes to be tested from multiple backgrounds.

"genes to be tested multiple genetic backgrounds" has been replace with "genes to be tested from multiple backgrounds” (line 396-397).

References:

Arndt D, Grant JR, Marcu A, Sajed T, Pon A, Liang Y, Wishart DS. 2016. PHASTER: a better, faster version of the PHAST phage search tool. Nucleic Acids Res 44:W16–W21.

Lawson DJ, Hellenthal G, Myers S, Falush D. 2012. Inference of Population Structure using Dense Haplotype Data. PLoS Genet 8:e1002453.

Liu M, Li X, Xie Y, Bi D, Sun J, Li J, Tai C, Deng Z, Ou H-Y. 2019. ICEberg 2.0: an updated database of bacterial integrative and conjugative elements. Nucleic Acids Res 47:D660–D665.

Parks DH, Imelfort M, Skennerton CT, Hugenholtz P, Tyson GW. 2015. CheckM: assessing the quality of microbial genomes recovered from isolates, single cells, and metagenomes. Genome Res 25:1043–1055.

Pascoe B, Méric G, Yahara K, Wimalarathna H, Murray S, Hitchings MD, Sproston EL, Carrillo CD, Taboada EN, Cooper KK, Huynh S, Cody AJ, Jolley KA, Maiden MCJ, McCarthy ND, Didelot X, Parker CT, Sheppard SK. 2017. Local genes for local bacteria: Evidence of allopatry in the genomes of transatlantic Campylobacter populations. Mol Ecol 26:4497–4508.

Robertson J, Nash JHE. 2018. MOB-suite: software tools for clustering, reconstruction and typing of plasmids from draft assemblies. Microb genomics 4.

Sheppard SK, Cheng L, Méric G, de Haan CPA, Llarena A-K, Marttinen P, Vidal A, Ridley A, Clifton-Hadley F, Connor TR, Strachan NJC, Forbes K, Colles FM, Jolley KA, Bentley SD, Maiden MCJ, Hänninen M-L, Parkhill J, Hanage WP, Corander J. 2014. Cryptic ecology among host generalist Campylobacter jejuni in domestic animals. Mol Ecol 23:2442–2451.

Sheppard SK, Colles F, Richardson J, Cody AJ, Elson R, Lawson A, Brick G, Meldrum R, Little CL, Owen RJ, Maiden MCJ, McCarthy ND. 2010. Host Association of Campylobacter Genotypes Transcends Geographic Variation. Appl Environ Microbiol 76:5269–5277.